TECHNICAL RELEASE

# A novel variable neighborhood search approach for cell clustering for spatial transcriptomics

Aleksandra Djordjevic[1,*,†], Junhua Li[1,2], Shuangsang Fang[1,3], Lei Cao[1,3] and Marija Ivanovic[1,*,†]

1 BGI Research, Shenzhen, 518083, China
2 BGI Research, Lidostas Parks, Riga, 49276, Latvia
3 BGI Research, Beijing, 102601, China

## ABSTRACT

This paper introduces a new approach to cell clustering using the Variable Neighborhood Search (VNS) metaheuristic. The purpose of this method is to cluster cells based on both gene expression and spatial coordinates. Initially, we confronted this clustering challenge as an Integer Linear Programming minimization problem. Our approach introduced a novel model based on the VNS technique, demonstrating the efficacy in navigating the complexities of cell clustering. Notably, our method extends beyond conventional cell-type clustering to spatial domain clustering. This adaptability enables our algorithm to orchestrate clusters based on information gleaned from gene expression matrices and spatial coordinates. Our validation showed the superior performance of our method when compared to existing techniques. Our approach advances current clustering methodologies and can potentially be applied to several fields, from biomedical research to spatial data analysis.

**Submitted:** 22 November 2023

\* Corresponding authors. E-mail: aleksandradradjordjevic@genomics.cn; marijaivanovic@genomics.cn

† Contributed equally.

Preprint submitted at https://doi.org/10.32388/0Z3EG4

Included in the series: *Spatial Omics: Methods and Application* (https://doi.org/10.46471/GIGABYTE_SERIES_0005)

**Subjects** Software and Workflows, Bioinformatics, Transcriptomics

## STATEMENT OF NEED

In high-throughput omics, deciphering the intricate cellular dynamics within tissues is pivotal [1, 2]. Cell clustering is essential for dissecting the mosaic of cellular diversity [3, 4]. This analytical approach seeks to categorize individual cells based on shared molecular signatures, allowing the identification of discrete subpopulations within heterogeneous tissues. In exploring cellular behavior and function, cell clustering emerges as an indispensable tool, providing insights into the subtle nuances of gene expression profiles. The ability to stratify cells into meaningful clusters not only refines our understanding of tissue composition but also lays the groundwork for precise insights into disease etiology and potential therapeutic interventions.

In tandem with cell clustering, spatial transcriptomics [5, 6] constitutes a revolutionary frontier for understanding cellular dynamics with their native microenvironments. Beyond the traditional scope of genomics, spatial transcriptomics integrates the spatial context of cells into the analysis, allowing researchers to explore how gene expression patterns unfold across complex tissue structures. This multidimensional approach surpasses the limitations of conventional transcriptomic studies, providing a spatially resolved perspective that is indispensable for decoding the orchestration of cellular interactions and the emergence of tissue-specific functions.

In order to contribute to this dynamic landscape, we introduce a novel methodology rooted in the Variable Neighborhood Search approach [7]. Our innovation seeks to elevate the precision and efficacy of cell clustering in spatial transcriptomic analyses, promising to reveal hidden facets of cellular organization and functionality. In this work, we introduce a novel Variable Neighborhood Search (VNS) approach tailored for cell clustering in spatial transcriptomics. Although our initial investigations focused on datasets designed for cell-type clustering, it is essential to emphasize that our method's design accommodates spatial domain clustering as well. Here, we present a synthesis of computational skills and biological insights aimed at pushing the boundaries of our understanding of the complex cell interactions within tissues.

## BACKGROUND

### Clustering methods from the literature

Many methods in the literature can be used to partition an *N*-dimensional population into *K* sets based on specific rules. In this paper, we focus on some of the most popular clustering methods used in the field of data analysis, such as *k*-Means [8], Louvain [9], Leiden [10], and MClust [11]. While these methods share the goal of grouping data points, they differ in the types of data they are designed for, the principle they optimize, and the algorithms they are well-suited for. *k*-Means is a general-purpose clustering algorithm, Louvain and Leiden are tailored for community detection in networks, while MClust is a model-based clustering method. In the following subsections, we briefly describe each of these methods.

#### *k-Means algorithm*

The *k*-means algorithm [8] is a partitioning algorithm that divides a dataset into *k*-clusters based on the similarity of data points. It starts by establishing *k* groups, each comprising a singular randomly chosen point. Points are then added to these groups according to the principle that new points are assigned to the group whose mean point is the most similar by some rule. After point allocation, the means of all groups are adjusted to incorporate the influence of newly added points. Consequently, at each stage, the *k*-means are reflective of the means of the groups they represent.

While this method is computationally efficient and adeptly handles extensive datasets, it does not guarantee convergence to an optimal solution. Notably, issues arise from the random initialization of centroids, leading to unexpected convergence patterns. Moreover, the algorithm requires users to choose the cluster number beforehand, influencing cluster shapes and susceptibility to outlier effects. However, it is known that certain special cases of the *k*-means algorithm exist in the literature where convergence to an optimal solution is assured.

#### *Louvain algorithm*

The Louvain algorithm, developed by Vondel *et al.* [9], is designed for detecting communities in network or graph data. This algorithm aims to optimize modularity, a measure of the quality of network division into communities, using two phases: (1) local moving of nodes and (2) aggregation of the network. In the first phase, individual nodes are moved to the community that yields the largest increase in the quality function. In the second phase, an aggregation network is obtained based on partitions, with each community in a partition becoming a node in the aggregate network. These two phases are



repeated until the quality function cannot be increased further. However, the Louvain algorithm can potentially produce communities with arbitrarily poor connectivity. In the most adverse scenarios, these communities may become entirely disconnected, particularly during iterative executions of the algorithm.

### Leiden algorithm

To address the connectivity issues of the Louvain algorithm, Traag *et al.* introduced the Leiden algorithm [10]. The Leiden algorithm guarantees that communities are well connected and, when applied iteratively, the algorithm converges to a partition where all subsets of all communities are locally optimally assigned. Leveraging the smart local move algorithm, an enhancement of the Louvain algorithm, and incorporating the concept of speeding up local node movements and moving nodes to random neighbours, the Leiden algorithm identifies the most promising directions for Louvain algorithm improvement. The Leiden algorithm consists of three phases: (1) local moving of nodes, (2) refinement of the partition, and (3) aggregation of the network based on the refined partition, using the non-refined partition to create an initial partition for the aggregate network. Thus, this algorithm optimizes a quality function to identify communities by considering the density of connections within the communities.

### MClust

MClust [11], applied in cell clustering, identifies distinct cell groups based on observed features using Gaussian mixture models [12]. Unlike other clustering algorithms, MClust accommodates various cluster shapes, making it suitable for complex situations. It utilizes the Expectation–Maximization [13] algorithm for parameter estimation, offering robust handling of missing data and complex distributions. This model-based clustering tool is powerful in uncovering patterns within complex biological datasets, such as those from single-cell omics technologies. Initially designed for single-cell RNA sequencing data, it can also be applied to spatial transcriptomic data, its effectiveness depending on data characteristics and analysis goals.

## Embedding methods from the literature

In spatial transcriptomics, where data is organized as a matrix with cells and genes, the high dimensionality (often exceeding 30,000 genes) and sparsity pose analytical challenges. Dimensionality reduction methods play key roles in addressing these issues. These techniques help distill meaningful patterns from the data, facilitating more efficient analyses.

The generation of embeddings, achieved through established literature methods, aims to transform the high-dimensional gene space into a more manageable form. This process enables a clearer exploration of spatial relationships, cell heterogeneity, and underlying biological processes. By leveraging validated methods from existing literature, we ensure a scientifically rigorous approach, condensing rich gene expression profiles into interpretable embeddings while addressing computational complexities.

As mentioned previously, we performed dimensionality reduction using five different embedding methods: STAGATE [14], Principal Component Analysis (PCA) [15], GraphST [16], Cell Clustering for Spatial Transcriptomics (CCST) data [17], and STAligner [18].

### STAGATE

The STAGATE method [14] has been designed for spatial clustering and denoising in spatially resolved transcriptomics data. This method generates low-dimensional latent embeddings with both spatial information and gene expressions via a graph attention auto-encoder. Notably, the method adopts an attention mechanism in the middle layer of the encoder and decoder, which learns the edge weights of spatial neighbor networks and uses them to update spot representations by collectively aggregating information from their neighbors.

### Principal Component Analysis

PCA [15] is a statistical method for dimensionality reduction and data visualization. It is a mathematical procedure that transforms a set of correlated variables into a new set of uncorrelated variables known as principal components. The principal components are linear combinations of the original variables and are sorted based on how much they account for the variance within the data; i.e., the first principal component accounts for the highest variance. PCA finds widespread application across domains, including data analysis, machine learning, and image processing, aiming to streamline intricate datasets and uncover patterns or associations between variables.

### GraphST

GraphST [16] is an advanced self-supervised contrastive learning technique designed to maximize the potential of spatial transcriptomics data. Integrating graph neural networks with self-supervised contrastive learning, this method acquires spot representations that are both informative and distinctive. This is achieved by minimizing the embedding distance between spatially neighboring spots reciprocally.

### Cell Clustering for Spatial Transcriptomics data

CCST [17] leverages graph convolutional networks (GCNs) to integrate gene expression data and comprehensive spatial information from individual cells in spatial gene expression data. The relationships between variables are captured as a graph, with the adjacency matrix representing connections among variables and the node feature matrix reflecting variable observations. The GCN layer is strategically designed to fuse graph (in our case, spatial structure) and node features (gene expression). Initially, the data is transformed into a graph, where nodes represent cells with gene expression profiles as attributes, and edges represent neighborhood relationships between cells. Subsequently, a sequence of GCN layers is used to incorporate graph and gene expression details into cell node embedding vectors. Concurrently, the graph is perturbed to generate negative embeddings. By learning the discrimination task, the neural network model is trained to encode cell embeddings derived from spatial gene expression data, subsequently used for cell clustering.

### STAligner

STAligner [18] is a specialized tool for aligning and integrating spatially-resolved transcriptomics data. It begins by normalizing expression profiles for all spots and creating a spatial neighbor network based on spatial coordinates. Employing a graph attention auto-encoder neural network, STAligner extracts spatially-aware embeddings and uses spot triplets to guide the alignment process, fostering similarity among related spots and



distinction among dissimilar ones across slices. The introduction of triplet loss refines spot embeddings by minimizing the distance from the anchor to positive spots and increasing the distance to negative spots. This iterative process optimizes triplet construction and auto-encoder training until batch-corrected embeddings are obtained. Furthermore, STAligner's versatility extends to integrating spatial transcriptomics datasets, facilitating alignment and concurrent identification of spatial domains across diverse biological samples, technological platforms, developmental stages, disease conditions, and consecutive tissue slices for 3D alignment.

## IMPLEMENTATION

### Mathematical model

Let $C = [c_i]$ represent the set of cells $c_i$, $i = 1, ..., n$, and the total number of cells be equal to $n$. For each cell $c_i$, $i = 1, ..., n$, let $c_i^x$ and $c_i^y$ represent its $x$ and $y$ coordinates, and let vector $c_i^{\text{emb}} = [c_i^{\text{emb}_1}, ..., c_i^{\text{emb}_M}]$ represent embedding values ($M$ is the total number of embedding values). Furthermore, let the distance function $D : C \times C \to \mathcal{R}^+$ be defined as a measure of the similarity between the cells. In our model, for two cells $c_i$ and $c_j$, the distance $D$ was calculated as follows: $D = a\, D_{\text{gene}} + (1 - a)D_{\text{coord}}$, where $a$ is the input parameter, $D_{\text{gene}}$ is the cosine similarity between cell embeddings, and $D_{\text{coord}}$ is the Euclidian distance between cell coordinates:

$$D_{\text{gene}}(c_i, c_j) = \text{cosine}(c_i, c_j),$$
$$D_{\text{coord}}(c_i, c_j) = \sqrt{(c_i^x - c_j^x)^2 + (c_i^y - c_j^y)^2}.$$

In our model, we chose $K$ different cells from the set of cells $C$ to represent clusters and called these cells *centroids*. Therefore, let the binary variables $x_{ij}$ ($i, j = 1, ..., n$) and $y_i$ be defined in the following way:

$$x_{ij} = \begin{cases} 1, & \text{if cell } c_i \text{ belongs to the cluster represented by centroid } c_j \\ 0, & \text{otherwise} \end{cases}$$

$$y_i = \begin{cases} 1, & \text{if cell } c_i \text{ represents the centorid} \\ 0, & \text{otherwise.} \end{cases}$$

The Integer Linear Programming formulation of the clustering problem can be described as follows:

$$\min \sum_{i=1}^{n} \sum_{j=1}^{n} x_{ij} D(c_i, c_j) \tag{1}$$

subject to these constraints:

$$\sum_{i=1}^{n} x_{ij} = 1, 1 \le j \le n, \tag{2}$$

$$x_{ij} \le y_j, 1 \le i \le n, 1 \le j \le n, \tag{3}$$

$$\sum_{i=1}^{n} y_i = K, \tag{4}$$

$$x_{ij}, y_j \in \{0, 1\}, 1 \le i \le n, 1 \le j \le n. \tag{5}$$



The objective function (1) represents the sum of distances from each cell to its most similar cluster representative. This function should be minimized. Equation (2) indicates that each cell is assigned to only one cluster. Before assigning a cell to a cluster, the cluster needs to be defined (3). The total number of clusters is equal to *K* (4). All variables are constrained to be binary (5).

The model described with Equations (1)–(5) is based on the *p*-median classification and is presented in a similar form by Davidović *et al.* [19].

## Variable neighborhood search method

The VNS method is a well-known metaheuristic method. It starts from one point in the search space, explores its neighborhoods, and repeats the process until a better solution or stopping criteria are reached. This method was proposed for the first time by Mladenović [20] and later elaborated by Mladenović and Hansen [21] and Hansen and Mladenović [22].

Before we introduce the VNS method, let us define the set $N_k(X), k = k_{\{min\}}, \ldots, k_{max}$ as the set of all vectors $X'$ that have a difference of the *k*th order from the solution *X*, and call that set *k*th Neighborhood to the solution *X*.

The VNS-based heuristic can be defined in a way that it starts from the initial feasible solution *X*, shakes it by creating another solution $X' \in N_k(X)$, and then applies a local search method to create a better feasible solution $X''$. If the feasible solution $X''$ obtained by the local search procedure is not better than the current incumbent *X* ($F(X'') \geq F^*$), the VNS algorithm repeats the procedure of shaking in the neighborhood $N_{k+k_{step}}$ (i.e., *k* is incremented by $k_{step}$) and local searches within it. It repeats this passage until *k* reaches its maximum $k_{max}$. Otherwise, if $F(X'') < F^*$, $F^*$ becomes $F(X'')$ and *k* becomes $k_{min}$. The procedure of changing the neighborhood enables the VNS algorithm to get out from the local minima. The process is repeated until a certain number of iterations or other stop criteria are reached.

Pseudo-code for the basic VNS algorithm is presented as Algorithm 1. Implementations of the functions *InitialSolution(), Shake(), LocalSearch(),* and *StoppingCondition()* defined for our clustering problem are described in the following subsection.

---

**Algorithm 1** (Basic) Variable Neighborhood Search Method

```
 1:  X* ← InitialSolution()
 2:  F* ← F(X*)
 3:  while StoppingCondition() do
 4:        k ← k_min
 5:        while k ≤ k_max do
 6:            X ← X*
 7:            X' ← Shake(X, k)
 8:            X'' ← LocalSearch(X')
 9:            if F(X'') < F* then
10:                X* ← Y''
11:                F* ← F(X*)
12:                k ← k_min
13:            else
14:                k ← k + k_step
15:            end if
16:        end while
17:  end while
```

---



## VNS for the cell clustering problem

With respect to the problem's definition, let us assume that all cells can be represented by numbers from 1 to *n*. Specifically, cells can be represented by the set $C = [c_i]$, $n = |C|$, and that for each cell $c_i$ there are two types of data: the *x* and *y* coordinates of the cell ($c_i^x$ and $c_i^y$) and the embedding values (vector $emb_i$). In our representation, the solution vector $Y = [y_1, ..., y_K]$ contains indexes of *K* cells chosen as cluster representatives. Also, cell $y_i$ is a centroid of the *i*-th cluster. From the centroid solution vector *Y* we obtain vector $X = [x_i]$ of size *n* in the following way: $x_i$, *i* = 1, ..., *n*, represents the closest centroid from the *Y* vector to the *i*-th cell. Our representation satisfies all conditions described by Equations (2)–(5). Using this representation, our goal was to minimize the value of the function $F : C \times C \to \mathcal{R}^+$, where *F* is defined as $F(X) = \sum_{i=1}^{n} (\alpha D_{\text{gene}}(i, x_i) + (1 - \alpha) D_{\text{coord}}(i, x_i))$.

The function *InitialSolution()* randomly chooses *K* mutually different numbers from the set of numbers $\{1, \ldots, n\}$ and returns them as a *K*-dimensional vector *Y*. For every solution vector *Y*, vector *X* is obtained in the following way: for each cell *i*, the distance *D* between the cell *i* and all centroids $y_j$ from the vector *Y* is calculated; next, $x_i$ is set equal to the $y_j$ for which the distance *D* is minimal. That is, whenever the vector *Y* is changed, vector *X* is also updated. Also, to avoid repeated calculations, the distance *D* between all cells is calculated and saved as a *distance* matrix.

The *Shake()* function takes two inputs: the incumbent *Y* and the size *k* of the neighborhood that needs to be explored. As a result, the *Shake()* function randomly chooses *k* elements from the vector *Y* and replaces them with *k* randomly chosen elements from the set $\{1, \ldots, n\}$ that are different from all elements from the current *Y*. This means that when some elements are changed, all elements in vector *Y* will still be mutually different. In other words, the *Shake()* function chooses a vector *Y′* from $N_k(Y)$.

The *LocalSearch* () function takes vector *Y′*, the distance matrix *distance*, and the parameters *m* and *p* as inputs. In our implementation, we used *the first improvement strategy*. Based on the value of the parameter *m*, for each element of the vector *Y′*, the *LocalSearch()* function first chooses a random integer number $ind \in [0, m]$; next, based on the *ind* value, keeps the observed element of the vector *Y′* as it is (*ind* == 0) or replace it with the new one (*ind* > 0). For *ind* ≥ 2, the observed element is replaced with one of the candidates from the set of candidates that are created within the *LocalSearch()* function (the *LocalSearch()* function searches for *ind* candidates for which the *distance* value from the observed candidate is the smallest, sorts the list, excludes all candidates that are already present in the vector *Y′*, and then chooses one candidate for the replacement). Please note that the smallest *distance* value between the observed candidate and itself will be zero, so the condition *ind* > 1 is necessary. In case *ind* == 1, *ind* will be chosen again until its value is not equal to 1. Additionally, if the candidate list is empty after excluding all elements that already exist in the vector *Y′*, a random candidate will be chosen from the set $\{1, \ldots, n\} \setminus \{y_1, \ldots, y_K\}$.

Finally, after the procedure of replacing or keeping elements from the vector *Y′* is finished, i.e., a new vector *Y″* is obtained, the *LocalSearch* () function calculates *F* (*Y″*) and, if *F* (*Y″*) < *F*\*, the first improvement has been made, and the function returns the vector *Y″* as the output or repeats the whole process. The process of examining elements of the vector *Y′* and replacing them with new values is repeated only if no improvement is made, but not more than *p* times. In case no improvement is made and the process has been repeated *p* times, the vector $Y'' = Y'$ will be returned as the output of this function.



In other words, the *LocalSearch()* function examines elements in the close neighborhood of the observed vector $Y'$ by creating a new vector $Y''$, calculates the function value $F(Y'')$ and, if the function value is less than the currently best value $F^*$, returns that vector. Otherwise, it will continue the process of examining elements of the vector $Y'$ but not more than $p$ times.

Usually, the *StoppingCondition()* function checks if the maximal number of iterations ($\max_{iter}$) or the maximal running time ($t_{max}$) have been reached. In our code, the *StoppingCondition()* function checks only if the maximal number of iterations has been reached and, if the answer is *true*, returns the best solution found as the result of the VNS procedure. If the maximal number of iterations has not been reached, the VNS procedure continues its search.

## DATA DESCRIPTION

We assessed the performance of the clustering methods through quantitative evaluation, employing datasets sourced from two distinct spatially resolved transcriptomic technologies: Stereo-seq [23] and 10x Visium [24].

From Stereo-seq technology, two datasets were used for testing: a large dataset of a field mouse brain hemisphere (**SS200000128TR E2 benchmark**) and another from the dorsal midbrain (**Forebrain**). The large field mouse brain contains more than 38,000 cells and more than 20,000 genes and can be downloaded from [25], while Forebrain contains more than 18,000 cells and more than 23,000 genes and can be downloaded from [26]. Please note that Forebrain contains the whole dorsal midbrain. In our study, we used manual lasso to separate a part of this dataset and called that part **Forebrain**. Both datasets are composed of only one slice.

In order to evaluate the performance of the presented VNS method on multi-slice datasets, we used a 10x Visium dataset containing spatial expressions of 12 human-layered dorsolateral prefrontal cortex (DLPFC) sections. Since these 12 sections are from three different human donors, they were used as multi-section (4-layers) datasets in our study. All layers of the DLPFC sections were manually annotated by Maynard *et al.* [24] and can be downloaded from [27]. Viewing them as the ground truth, we compared the clustering accuracy of the VNS method with other clustering methods using only embedding obtained by the vertical spatial transcriptomic integration provided by STAGATE.

## ANALYSIS

### Input parameters

Testing was conducted on the AWS cloud instance **m6a.48xlarge** under the Linux operative system.

Input parameters for our algorithm are the number of clusters ($K$), the percentage of the influence of the embedding values ($a$), the maximal number of neighborhoods that should be searched ($k_{max}$), the maximal number of iterations ($max_{iter}$), and the *local search* parameters $m$ and $p$. The minimal ($k_{min}$) number of neighborhoods and step ($k_{step}$) are set to 1 by default.

The input parameters used for testing are $\alpha \in \{1, 0.95\}$ ($a = 1$ means that no additional spatial information is included, while $a = 0.95$ means that 5% of spatial information is used for calculating the distance between the cells), $k_{max} \in \{10, 15, 20, 25, 30\}$, $m \in \{10, 12, 15, 20, 30\}$, and $p \in \{10, 12, 15, 20\}$.



**Table 1.** VNS solution for single-slice datasets. Values in columns $f_{VNS}$, $t_{VNS}$, $err$ and $\sigma$ are obtained as explained in the Analysis section.

| Embedding | $f_{VNS}$ | $t_{VNS}$ (s) | $err$ | $\sigma$ |
|---|---|---|---|---|
| | | E2 | | |
| CCST | 1,019.7419 | 48.8355 | 0.1626 | 0.0476 |
| STAGATE | 2,706.7446 | 110.258 | 0.1196 | 0.0415 |
| PCA | 9,550.0142 | 79.1977 | 0.0320 | 0.0118 |
| GraphST | 10,083.5379 | 64.95 | 0.0197 | 0.0059 |
| | | Forebrain | | |
| CCST | 427.8511 | 47.8054 | 0.1579 | 0.0439 |
| STAGATE | 543.0947 | 52.7096 | 0.0925 | 0.0347 |
| PCA | 3,541.7886 | 50.1935 | 0.0214 | 0.0073 |
| GraphST | 2,209.235 | 92.0103 | 0.0473 | 0.0140 |

## Evaluation method

We used the Adjusted Rand Index (ARI) [28] to evaluate the results and compare them with each other. ARI is a measure used to evaluate the performance and similarity between two clustering algorithms. It quantifies the agreement between the true and predicted clustering, adjusting for the amount of agreement that could occur by chance. ARI values range from −1 to 1: where 1 indicates the perfect agreement, 0 indicates agreement expected by chance, and negative values suggest less agreement than expected by chance.

## Results of the VNS method across various scenarios with single-slice datasets

Due to the sparsity of the gene expression matrix and to ensure a fair comparison, embeddings were obtained using various methods from the literature (PCA, STAGATE, GraphST, and CCST) for both Stereo-seq datasets. Moreover, all methods create embedding that significantly reduces the number of genes to a much smaller set of features. For instance, the CCST method reduced the number of genes from the Forebrain dataset to 128 features, STAGATE to 64 features, PCA to 50 features, and GraphST to 20 features. For the E2 dataset, all parameters were the same except for STAGATE, where the number of features was lowered to 30. Hence, the input data depend on the number of cells and the number of obtained features (embeddings). The standard clustering methods from the literature (*k*-Means, MClust, Louvain, and Leiden) and the proposed VNS method for cell clustering were applied to the generated embeddings. The results of the testing are presented in Tables 1 and 2.

The goal of the VNS method was to find the solution with the smallest cost function, and we show these results in Table 1. Table 1 shows results obtained by the VNS method only and is organized as follows: the first column presents the name of the embeddings used as the input to the VNS method, while the following four columns ($f_{VNS}$, $t_{VNS}$, $err$, and $\sigma$) show the smallest cost function value, the corresponding running time, and the statistical analysis of all solutions obtained by VNS when comparing to the presented cost function value in that order. In other words, due to the stochastic nature of the metaheuristic, the VNS algorithm was run 20 times (for 20 different seeds) for each embedding, and information regarding the best solution value obtained in these 20 runs is provided in these four columns ($f_{VNS}$, $t_{VNS}$, $err$, and $\sigma$). More precisely, $f_{VNS}$ presents the minimal cost function value obtained after these 20 runs; $t_{VNS}$ is the corresponding running time for the presented solution value; $err$ and $\sigma$ contain additional information on the quality of the



**Table 2.** Clustering method comparison for single-slice datasets. The highest *ARI* score achieved for some datasets among all clustering methods is highlighted in bold, while the second-best *ARI* score is highlighted by an asterisk (*).

| Embeddings | Leiden | | Louvain | | *k*-Means | | MClust | | VNS | |
|---|---|---|---|---|---|---|---|---|---|---|
| | *ARI* | *t* (s) | *ARI* | *t* (s) | *ARI* | *t* (s) | *ARI* | *t* (s) | *ARI* | *t* (s) |
| E2 | | | | | | | | | | |
| CCST | 0.1553 | 29.1638 | 0.1518 | 5.7702 | 0.1962* | 15.3243 | 0.1401 | 4,799.5287 | **0.2224** | 47.5667 |
| STAGATE | 0.1951 | 7.5198 | 0.2176 | 6.3803 | **0.2907** | 2.62854 | 0.2052 | 516.8929 | 0.2890* | 59.7737 |
| PCA | 0.0001 | 6.8347 | 0.1316 | 9.9780 | 0.2072* | 12.0037 | 0.2024 | 1,128.1911 | **0.2907** | 235.465 |
| GraphST | **0.0841** | 14.8255 | 0.0697* | 13.0344 | 0.0492 | 4.2599 | 0.0635 | 533.1441 | 0.0636 | 47.5184 |
| Forebrain | | | | | | | | | | |
| CCST | 0.0925 | 25.7164 | 0.0961* | 2.5659 | 0.1093 | 8.7788 | 0.0821 | 1,330.3455 | **0.1263** | 18.6987 |
| STAGATE | 0.1753 | 3.6952 | 0.1676 | 3.6263 | 0.1775* | 6.0085 | 0.1718 | 269.9742 | **0.2342** | 24.6907 |
| PCA | 0.1659 | 4.4805 | 0.1674* | 3.7720 | **0.1717** | 6.4302 | 0.1025 | 147.4443 | 0.1568 | 45.2866 |
| GraphST | 0.1738 | 3.8813 | 0.1847* | 4.6558 | 0.1833 | 1.8972 | 0.1709 | 73.0143 | **0.2104** | 9.2064 |

solution: *err* is the average relative error of found solution from the presented one and is calculated as $err = \frac{1}{20} \sum_{i=1}^{20} err_i$, where *err$_i$* = $|\text{VNS}_i - f_{\text{VNS}}| / |\text{VNS}_i|$, where VNS$_i$ is the VNS solution obtained in the *i*th run (seed). The value $\sigma$ is the standard deviation of *err* and is calculated by $\sigma = \sqrt{\frac{1}{20} \sum_{i=1}^{20} (err_i - err)^2}$. For each embedding method, the results obtained by VNS are presented in separate rows.

The results presented in Table 2 are organized into three groups. Similar to Table 1, the first column (first group) presents the name of the method used for creating the embedding. The next ten rows present the results for each clustering method separately; for each method, we provide the ARI score (*ARI*) and the running time (*t*) in seconds. The *ARI* and *t* values under the VNS columns stand for the best found *ARI* score obtained for all testing combinations and the corresponding running time. The highest *ARI* score achieved for some datasets among all clustering methods is highlighted in bold, while the second-best *ARI* score is highlighted by an asterisk (*).

In both tables, the first set of results corresponds to the E2 dataset, and the next corresponds to the Forebrain dataset. The E2 dataset results are visualized in Figure 1, while the Forebrain dataset results are visualized in Figure 2.

## VNS clustering achieves better results than other tested methods using the E2 dataset

From the first part of the results shown in Table 1, we can conclude that, using PCA embedding in all 20 runs, the values of the cost function are very close to the lowest cost function value (*err* < 3.5, $\sigma$ < 1.5%). Using STAGATE, we have some differences, although $\sigma$ is still below 5% implying that the VNS method is stable with both embeddings. The results of VNS clustering when the smallest cost function values are reached are visualized in Figure 1a, while the results with the best ARI score achieved by all clustering methods are shown in Figure 1b.

## VNS methods outperform other methods when clustering cells from the Forebrain dataset

By examining values from the *err* and $\sigma$ columns in Table 1 for the Forebrain dataset, it can be easily seen that differences between the results obtained in 20 runs are very small. In fact, the difference between the best-found solution (the solution with the minimal cost function value) and the other 19 solutions is less than 5% (the average relative error $\sigma$ is

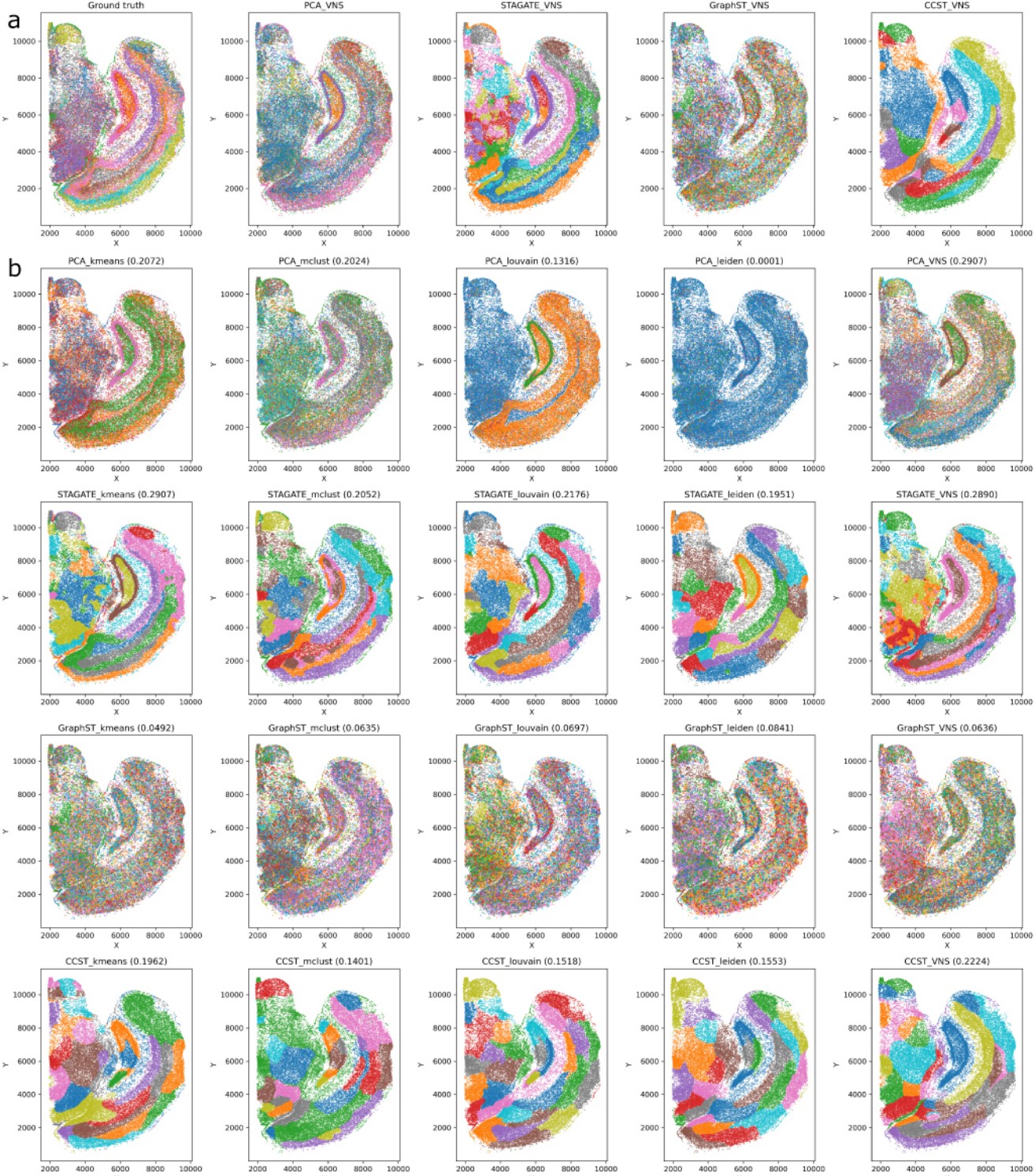

**Figure 1.** (a) Results of the VNS clustering on the E2 dataset. The first figure on the left presents the ground truth data. These results were obtained using the VNS method with PCA, STAGATE, GraphST, and CCST embeddings. (b) Clustering results for the E2 dataset. Each row presents the clustering results obtained by *k*-Means, MClust, Louvain, Leiden, and VNS over a certain embedding method. Therefore, the first row presents the results obtained by all clustering methods when using PCA embedding. The next three rows used STAGATE, GraphST, and CCST embeddings.

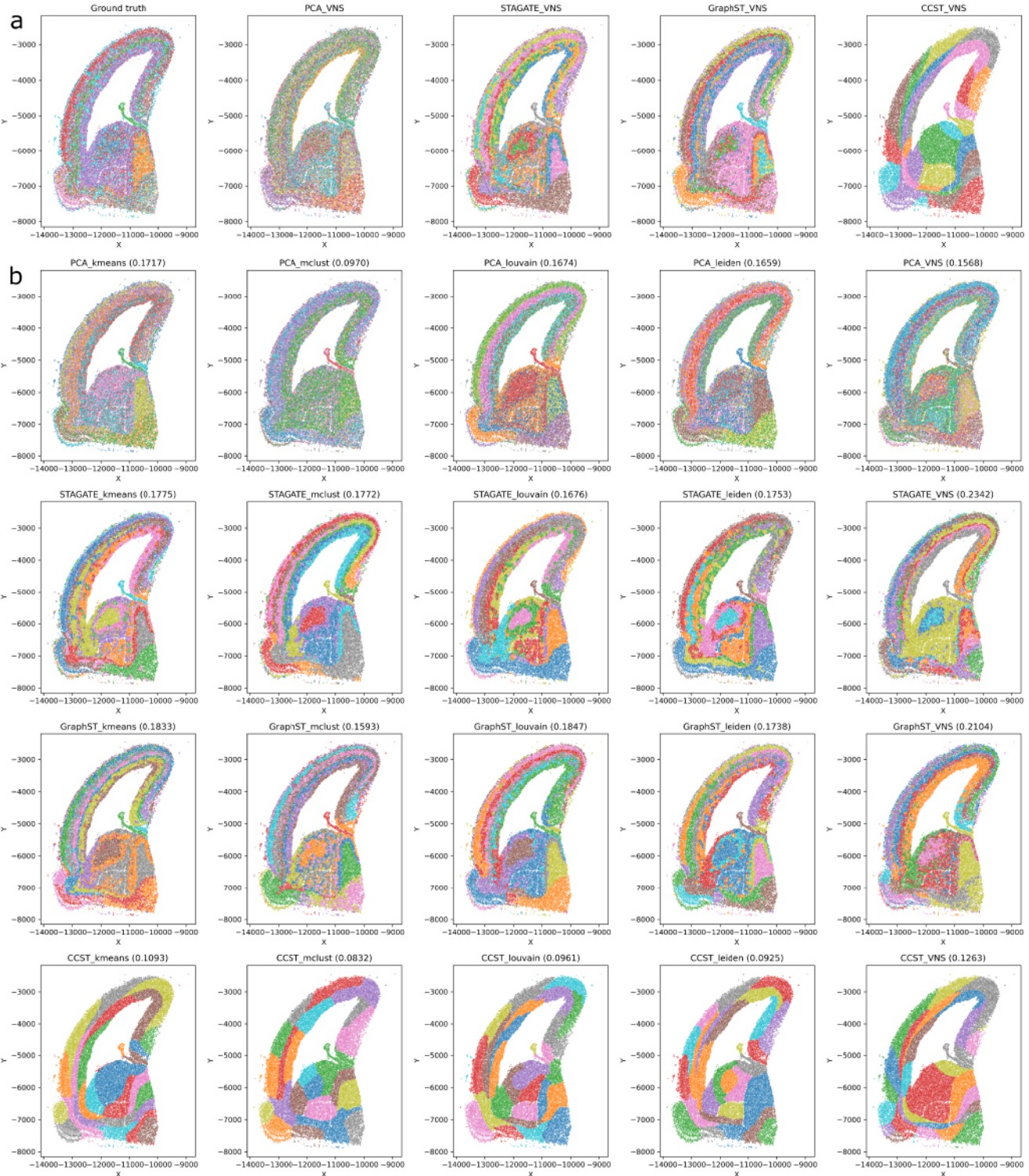

**Figure 2.** (a) Results of the VNS clustering on the Forebrain dataset. The first figure on the left presents the ground truth data. These results were obtained using the VNS method with PCA, STAGATE, GraphST, and CCST embeddings. (b) Clustering results for the Forebrain dataset. Each row presents the clustering results obtained by *k*-Means, MClust, Louvain, Leiden, and VNS, over a certain embedding method. Therefore, the first row presents the results obtained by all clustering methods when using PCA embedding. The next three rows used STAGATE, GraphST, and CCST embeddings.

**Table 3.** VNS solution for multi-slice datasets.

| Slice name | $f_{VNS}$ | $t_{VNS}$ | *err* | $\sigma$ |
|---|---|---|---|---|
| 151507_151510 | 890.7088 | 4.2262 | 0.0884 | 0.0390 |
| 151669_151672 | 755.7133 | 2.8674 | 0.0866 | 0.0273 |
| 151673_151676 | 513.8781 | 1.1983 | 0.0923 | 0.0396 |

**Table 4.** Clustering method comparison for multi-slice datasets. The highest *ARI* score achieved for some datasets among all clustering methods is highlighted in bold, while the second-best *ARI* score is highlighted by an asterisk (*).

| Slice name | Leiden | | Louvain | | *k*-Means | | MClust | | VNS | |
|---|---|---|---|---|---|---|---|---|---|---|
| | *ARI* | *t* (s) | *ARI* | *t* (s) | *ARI* | *t* (s) | *ARI* | *t* (s) | *ARI* | *t* (s) |
| 151507_151510 | 0.3440 | 27.3778 | 0.4293* | 4.0119 | 0.3061 | 2.1001 | 0.3489 | 62.5176 | **0.4887** | 2.1094 |
| 151669_151672 | 0.4084 | 26.9197 | 0.4985* | 2.9611 | 0.2213 | 1.6839 | 0.4633 | 39.1007 | **0.6156** | 1.3014 |
| 151673_151676 | 0.4370 | 25.1056 | 0.4754* | 2.6766 | 0.3299 | 1.4413 | 0.4316 | 49.1890 | **0.5016** | 0.8573 |

less than 5%). This result means that the solutions found in all 20 runs were very close to the smallest one. Also, from the results in the column $t_{VNS}$, we can observe a running was less than 1 minute for three different embedding types and less than 2 minutes for one embedding type.

Moreover, from the results presented in Table 2 for the Forebrain dataset, we can see that, in the majority of cases, VNS had the highest *ARI* score compared to the other methods (for three types of embedding, the *VNS ARI* score was the highest). Also, the running time was less than 1 minute for each type of embedding. The only embedding for which the VNS did not find a solution with the best *ARI* score was the PCA one, and for this embedding, the best *ARI* score was obtained by the *k*-Means method.

By analyzing the results in Tables 1 and 2, we conclude that the VNS method achieves the best *ARI* score with the STAGATE embedding, and that in all 20 runs all solutions were close to the one with the lowest cost function (*err* < 1%). The results obtained with the minimal cost function and the maximal *ARI* score are visualized in Figure 2.

## VNS demonstrates a superior performance on multi-slice datasets

Next, we compared the clustering accuracy of the VNS method with other clustering methods by using embeddings obtained by the STAligner method only. Compared to other embedding methods used for single-slice datasets, it is worth mentioning that STAligner reduces the number of genes to 30 features. The results of this comparison are presented in Tables 3 and 4. Table 3 is organized similarly to Table 1. The only difference is in the first column, which, in this case, is called Slice name. Since DLPFC datasets are 4-layered slices, this column contains the names of the first and the last slices in this particular dataset. The same case applies to other slices. Thus, each row represents the results for one separate DLPFC dataset.

Table 4 is organized similarly to Table 2; however, the column Embeddings is replaced by the column Slice name, and the names of the first and the last slices from particular multi-slice datasets are presented. And again the same case applies to other slices. The results for each dataset are presented in separate rows, as in Table 3. The results from Table 3 are visualized in Figure 3.

As we see from the columns *err* and $\sigma$ in Table 3, in all 20 runs, the VNS method obtained results similar to the ones with the smallest cost function (*err* < 5.8%, $\sigma$ < 2.5%). Again, these results imply that the method is stable even for multi-slice datasets. The fact that results

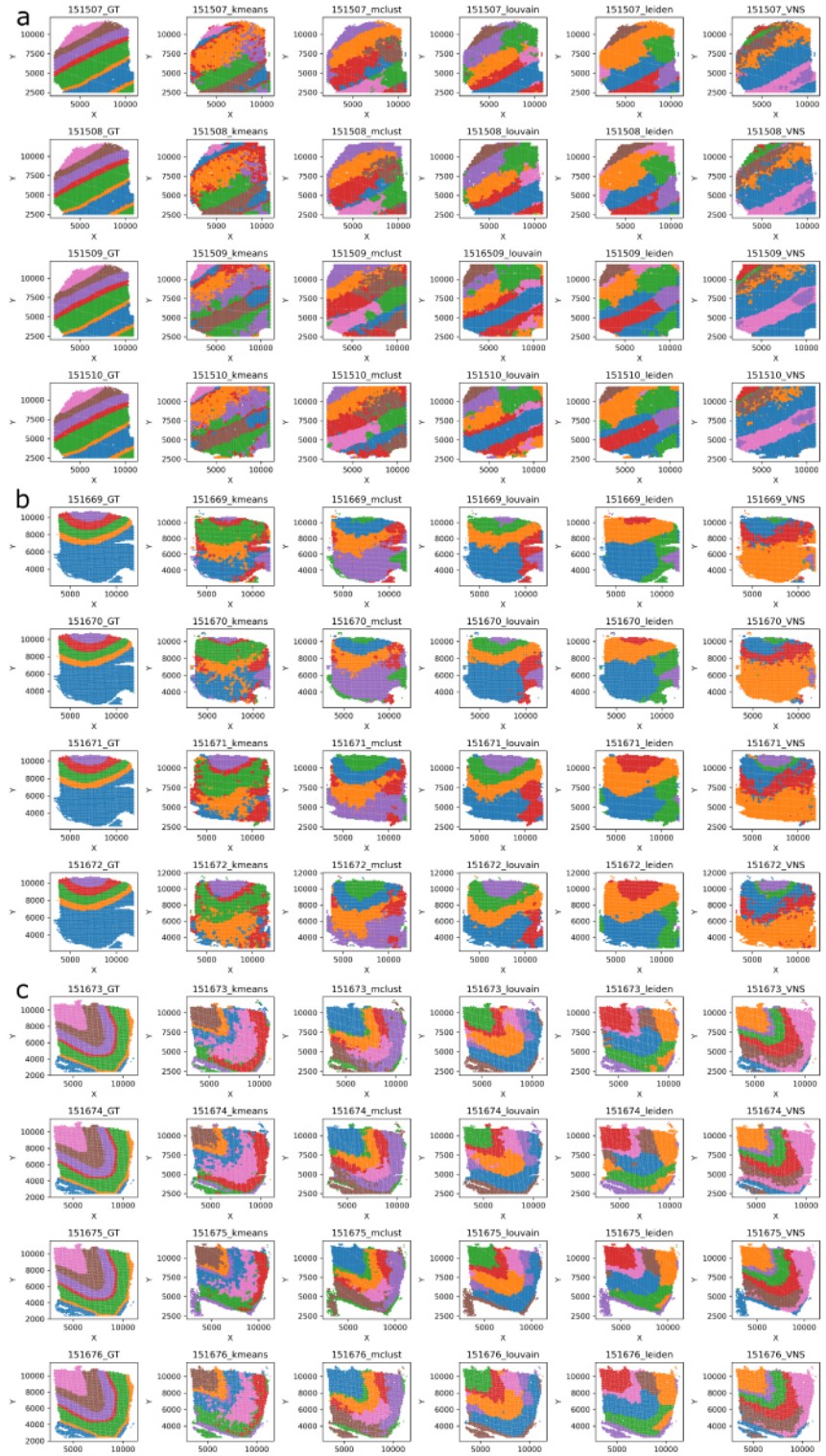

**Figure 3.** The clustering results on the DLPFC datasets 151507-151510, 151669-151672, and 151673-151676 are presented in panels (a), (b), and (c), respectively. The first column shows the ground truth data, while the subsequent columns display the results obtained using *k*-Means, MClust, Louvain, Leiden, and the VNS method with STAligner embeddings.

from the columns $t_{\text{VNS}}$ are smaller than 5 implies that this method can obtain results for four slices of these types of datasets in less than 5 s.

From the results presented in Table 4, it can be concluded that the method proposed in this paper outperforms other clustering methods in all aspects. Specifically, for each of the datasets we tested, *ARI* score was the highest and the running time was the lowest when the VNS method was used.

## DISCUSSION AND CONCLUSION

Here, we introduced a novel approach suitable for clustering both single- and multi-slice spatial transcriptomics datasets. This is the first application of a metaheuristic method, called the VNS, to the clustering of spatial transcriptomic data. The essence of the VNS implementation presented in this study is the utilization of a combinatorial/mathematical optimization algorithm; in this instance, a metaheuristic approach. These methods are strategically designed to deliver sufficiently optimal solutions to optimization and machine learning challenges while minimizing computational resources. This approach is intended to offer a robust and computationally efficient solution for cell clustering in spatial transcriptomics.

Our analysis demonstrated that the performance of clustering methods is significantly influenced by the choice of embeddings and the way they were generated. Notably, the VNS approach combined with PCA embeddings yields results that closely align with the ground truth, as illustrated in Figure 2b. When benchmarked against existing techniques, our method consistently outperforms in terms of efficiency and ARI scores. The algorithm's speed and stability are commendable, and its flexibility is evidenced by a comprehensive set of parameters that can be tailored to meet diverse user requirements. Future research will extend the method's application to time-series datasets and explore additional VNS modifications and embedding techniques to enhance its utility.

## AVAILABILITY OF SOURCE CODE AND REQUIREMENTS

- Project name: VNS
- Project home page: https://github.com/STOmics/VNS/tree/main
- Operating system(s): Linux
- Programming language: Python
- License: MIT
- RRID:SCR_024993

## DATA AVAILABILITY

From Stereo-seq technology, two datasets were used:

(1) a large dataset of a field mouse brain hemisphere (**SS200000128TR E2 benchmark**), which can be downloaded from Zenodo [25].

(2) **Forebrain**, which can be downloaded from the CNGB MOSTA database https://db.cngb.org/stomics/mosta/download/.

We used only one part of Forebrain, which was extracted using a manual lasso. Additional data is also available in GigaDB [29] and snapshots of the code are archived in Software Heritage [30] under the implementation part as it is shown in Figure 4.

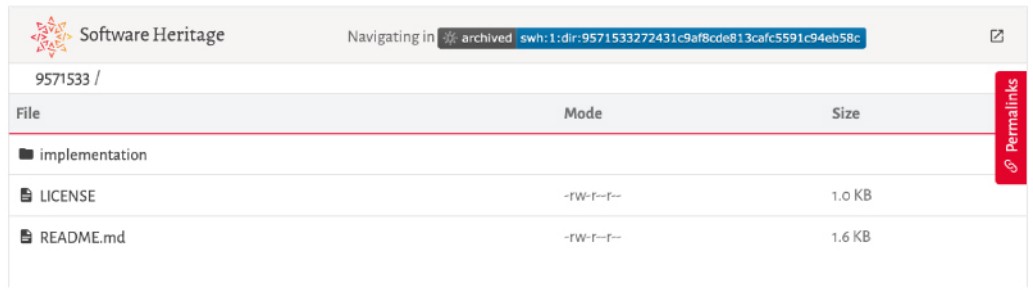

**Figure 4.** Software Heritage archive of the code [30]. https://archive.softwareheritage.org/browse/embed/swh:
1:dir:9571533272431c9af8cde813cafc5591c94eb58c;origin=https://github.com/STOmics/VNS;visit=swh:1:snp:
98f7a1f57b2f1b8875a892c2666de615bf92d894;anchor=swh:1:rev:6b12ba4bfd7e7b6f273d2dfb25e55c8fb320d2ea/

## ABBREVIATIONS

ARI, Adjusted Rand Index; CCST, Clustering for Spatial Transcriptomics; CNGB, China National GeneBank; DLPFC, dorsolateral prefrontal cortex; GCN, graph convolutional network; PCA, Principal Component Analysis; VNS, Variable Neighborhood Search.

## DECLARATIONS

### Ethics approval and consent to participate

The authors declare that ethical approval was not required for this type of research.

### Competing interests

The author(s) declare that they have no competing interests.

### Consent for publication

Not applicable.

### Author's Contributions

AD and MI provided the idea of the solution, implementation, testing, and manuscript. JL and SF supervised the whole process. CL created embeddings for both datasets for testing.

### Funding

This work was supported by the National Key R&D Program of China (2022YFC3400400).

### Acknowledgements

We acknowledge the CNGB Nucleotide Sequence Archive (CNSA) of China National GeneBank DataBase (CNGBdb) for maintaining the MOSTA database.

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
