## [Reviewer Report]

Comments on revised manuscriptAuthors have made corresponding revisions according to my suggestions, and I accept this paper for publication in this journal.

---

## [Editor Report]

Editor’s AssessmentThis paper describes a new spatial transcriptomics method, presenting a new approach to cell clustering using the Variable Neighborhood Search (VNS) metaheuristic. Which clusters cells based on both gene expression and spatial coordinates. Being able to stratify cells into meaningful clusters helps the understanding of tissue composition and also provides insights into disease etiology and potential therapeutic interventions. Validating this new method demonstrated that the performance of clustering methods is significantly influenced by the choice of embeddings and the way they were generated. The implementation and description was very complicated, but after some review and editing the writing has been simplified and more case studies added. Future research is needed to extend the method’s application to time-series datasets and explore additional VNS modifications and embedding techniques to enhance its utility.

---

## [Reviewer Report]

Reviewer name and names of any other individual's who aided in reviewerXianwen RenDo you understand and agree to our policy of having open and named reviews, and having your review included with the published manuscript. (If no, please inform the editor that you cannot review this manuscript.)YesIs the language of sufficient quality?YesPlease add additional comments on language quality to clarify if neededIs there a clear statement of need explaining what problems the software is designed to solve and who the target audience is? YesAdditional CommentsIs the source code available, and has an appropriate Open Source Initiative license <a href="https://opensource.org/licenses" target="_blank">(https://opensource.org/licenses)</a> been assigned to the code?YesAdditional CommentsAs Open Source Software are there guidelines on how to contribute, report issues or seek support on the code?YesAdditional CommentsIs the code executable?YesAdditional CommentsIs installation/deployment sufficiently outlined in the paper and documentation, and does it proceed as outlined?YesAdditional CommentsIs the documentation provided clear and user friendly?YesAdditional CommentsAdditional CommentsIs there a clearly-stated list of dependencies, and is the core functionality of the software documented to a satisfactory level?YesAdditional CommentsHave any claims of performance been sufficiently tested and compared to other commonly-used packages? YesAdditional CommentsIs test data available, either included with the submission or openly available via cited third party sources (e.g. accession numbers, data DOIs)?YesAdditional CommentsAre there (ideally real world) examples demonstrating use of the software? YesAdditional CommentsIs automated testing used or are there manual steps described so that the functionality of the software can be verified?NoAdditional CommentsAny Additional Overall Comments to the AuthorThis manuscript presents a clustering algorithm that employs variable neighborhood search and integer linear programming. Benchmark on different datasets confirms the advantage regarding clustering accuracy and computational speed. Overall, it is elegantly designed and well-implemented. A minor error should be corrected before publication. On page 6, cosine is called a distance metric, which is wrong. Cosine is a similarity metric, not a distance metric.RecommendationMinor Revisions

---

## [Reviewer Report]

Upload additional filesTRR-202311-02/form/Reviewer Report.pdfReviewer name and names of any other individual's who aided in reviewerLi ZhangDo you understand and agree to our policy of having open and named reviews, and having your review included with the published manuscript. (If no, please inform the editor that you cannot review this manuscript.)YesIs the language of sufficient quality?YesPlease add additional comments on language quality to clarify if neededIs there a clear statement of need explaining what problems the software is designed to solve and who the target audience is? NoAdditional CommentsIs the source code available, and has an appropriate Open Source Initiative license <a href="https://opensource.org/licenses" target="_blank">(https://opensource.org/licenses)</a> been assigned to the code?YesAdditional CommentsAs Open Source Software are there guidelines on how to contribute, report issues or seek support on the code?YesAdditional CommentsIs the code executable?YesAdditional CommentsIs installation/deployment sufficiently outlined in the paper and documentation, and does it proceed as outlined?YesAdditional CommentsIs the documentation provided clear and user friendly?YesAdditional CommentsIs there enough clear information in the documentation to install, run and test this tool, including information on where to seek help if required?YesAdditional CommentsIs there a clearly-stated list of dependencies, and is the core functionality of the software documented to a satisfactory level?YesAdditional CommentsHave any claims of performance been sufficiently tested and compared to other commonly-used packages? YesAdditional CommentsIs test data available, either included with the submission or openly available via cited third party sources (e.g. accession numbers, data DOIs)?YesAdditional CommentsAre there (ideally real world) examples demonstrating use of the software? YesAdditional CommentsIs automated testing used or are there manual steps described so that the functionality of the software can be verified?NoAdditional CommentsAny Additional Overall Comments to the AuthorThis is an impressive study, and the author have presented a groundbreaking strategy leveraging the Variable Neighborhood Search (VNS) metaheuristic. However, I strongly believe that this paper could be accepted for publication in GigaByte after undergoing a complete revision addressing the following suggestions. 1. In this paper, the figures should be improved and the formulas should be more standardized. 2. In my opinion, it would be more beneficial for readers if the innovations of this manuscript are presented in a clearer manner. 3. I would like to see if the attention mechanism (transformer) works in your novel approach. I strongly believe that you would learn a lot about the transformer from these references (PMID: 36411674, PMID: 37788507). 4. You should conduct additional case studies to further validate the applicability of your method.RecommendationMajor Revisions